# Insights into the Impact of Hesitancy on Cancer Care and COVID-19

**DOI:** 10.3390/cancers15123115

**Published:** 2023-06-08

**Authors:** Nathan Visweshwar, Juan Felipe Rico, Irmel Ayala, Michael Jaglal, Damian A. Laber, Mohammad Ammad-ud-din, Lubomir Sokol, Eduardo Sotomayor, Arumugam Manoharan

**Affiliations:** 1Department of Medicine, University of South Florida, Tampa, FL 33612, USA; 2Department of Pediatric Hematology, University of South Florida, Tampa, FL 33612, USA; jrico@usf.edu; 3Department of Pediatric Hematology, Johns Hopkins All Children’s Hospital, St. Petersburg, FL 33701, USA; irmel.ayala@jhmi.edu; 4Department of Satellite and Community Oncology and Hematology, Moffitt Cancer Center, Tampa, FL 33612, USA; michael.jaglal@moffitt.org; 5FACP Department of Satellite and Community Oncology, Moffitt Cancer Center, Tampa, FL 33612, USA; damian.laber@moffitt.org; 6Department of Hematology and Oncology, Moffitt Cancer Center, Tampa, FL 33612, USA; mohammad.ammad-ud-din@moffitt.org; 7Department of Malignant Hematology, Moffitt Cancer Center, Tampa, FL 33612, USA; lubomir.sokol@moffitt.org; 8Malignant Oncology, Tampa General Hospital, Tampa, FL 33607, USA; 9FRACP, FRCPA Faculty of Science, Medicine and Health, University of Wollongong, Wollongong, NSW 2217, Australia; arumugam@uow.edu.au

**Keywords:** COVID-19, cancer, hesitancy, management, treatment

## Abstract

**Simple Summary:**

The impact of COVID-19 on cancer diagnosis and management during the pandemic remains uncertain. The risk of infection and travel restrictions made some cancer patients reluctant or unable to travel, hindering their access to diagnostic procedures and treatment. The pandemic also led to a shortage of health-care personnel, forcing cancer centers to limit or postpone screening procedures, such as mammography and colonoscopy, as well as elective surgeries, chemotherapy, and radiation therapy. Ophthalmic and ENT procedures, endoscopy, intubation, surgery, and chemotherapy infusions were also canceled to minimize close contact during the pandemic. This review will explore how delays in seeking medical care and reluctance to receive COVID-19 vaccinations have affected cancer patients.

**Abstract:**

World Health Organization findings indicate that the COVID-19 pandemic adversely affected cancer diagnosis and management. The COVID-19 pandemic disrupted the optimal management of outpatient appointments, scheduled treatments, and hospitalizations for cancer patients because of hesitancy among patients and health-care providers. Travel restrictions and other factors likely affected medical, surgical, and radiation treatments during the COVID-19 pandemic. Cancer patients were more likely to be affected by severe illness and complications if they contracted COVID-19. A compromised immune system and comorbidities in cancer patients may have contributed to this increased risk. Hesitancy or reluctance to receive appropriate therapy or vaccination advice might have played a major role for cancer patients, resulting in health-care deficits. The purpose of this review is to evaluate the impact of COVID-19 on screening, entry into clinical trials, and hesitancy among patients and health-care professionals, limiting adjuvant and metastatic cancer treatment.

## 1. Introduction

COVID-19 was classified as a global pandemic by the World Health Organization in March 2020.

The pandemic has significantly impacted cancer patients. There is an estimated two-fold increase in the risk of contracting COVID-19 in cancer patients compared to the general population. Furthermore, they have a 3.5-fold higher risk of developing severe COVID-19, leading to increased morbidity and mortality. These heightened risks are compounded by factors such as aging and the presence of comorbidities among cancer patients [1,2]. 

Just when the COVID-19 pandemic was thought to be under control, a developed country such as Japan, despite its well-organized national vaccination program, reported a cumulative total of 16,423,053 infected individuals and 36,234 deaths as of August 19, 2022. This was mainly from the Omicron variant of COVID-19, which led to the seventh wave of the pandemic and the statewide restriction of the movement of citizens in Japan—see Figure 1 [3].

As the original strain of the virus has receded, the recurrent and ever-evolving nature of viral infections has led to the emergence of virulent strains such as the Alpha, Delta, and Omicron variants. These new strains have been infecting individuals, including those who have already been vaccinated. A once-a-year COVID-19 vaccine has been proposed by the Infectious Diseases Society of America and the Food and Drug Administration, which underscores the potential persistence of the virus for years to come [4]. In this setting, continuing hesitancy with regard to COVID-19 vaccination, even in developed countries such as the United States with a 64.6% rate and in Russia with a 30.4% rate, remains concerning [5]. Unvaccinated (30 to 60%) individuals feel that herd immunity may protect them [5] (Figure 2). Thus far, there is no strategy to optimize defenses against the emergence of new COVID-19 variants [6].

In the early months and years of the COVID-19 pandemic, the complex health-care landscape contributed to fewer cancer diagnoses, as initial investigations of symptoms that could confirm cancer were delayed due to fear of the virus [7]. Screening programs for breast, colorectal, and cervical cancers were put on hold or suspended as the health-care system faced unprecedented demands in managing the pandemic [8].

Both patients and health-care providers during the COVID-19 pandemic postponed appointments at cancer centers, leading to missed diagnoses and treatment opportunities for cancer patients.

Health-care deficits during the COVID-19 pandemic were largely caused by hesitancy or reluctance. This delayed cancer diagnosis and treatment. The hesitancy was not only from patients avoiding travel to cancer centers because of the travel restrictions due to “lockdown” during the global pandemic but also hesitancy from physicians in seeing patients in health-care facilities. This was either because the clinicians were working remotely or because the health-care providers were overwhelmed with taking care of COVID-19 victims. This resulted in delayed clinic visits, cancellation of preventive measures, including mammography, colonoscopy, and PSA screening, avoidance of endoscopy and ENT/ophthalmological procedures and disruptions to surgery, chemotherapy and radiation therapy. During the COVID-19 pandemic, the pharmaceutical industry and other agencies responded by restricting/closing drug trials, either because of financial constraints or because of a lack of personnel to monitor these programs. Finally, when the COVID-19 vaccines were freely available, the hesitancy of cancer patients and providers prevented the COVID-19 vaccination of cancer patients because of an assumed lack of response from the underlying immunocompromised state, presumed lack of response from chemotherapy or fear of the detrimental effect of the vaccine (venous and arterial thromboembolism). This further restricted cancer patients’ access to life-saving therapy. This review will explore how delays in seeking medical care and reluctance to receive COVID-19 vaccinations have affected cancer patients.

## 2. Data Collection

For this narrative review, we used databases, including Medline and Embase, and guidelines from the American Society of Clinical Oncology and the European Society of Medical Oncology. We reviewed the literature for the impact of COVID-19 on cancer from January 2020 through December 2022. We also searched the World Health Organization, International Monetary Fund, COVAX Global Vaccine Assessment, the United Nations Children’s Fund (UNICEF), Duke Global Health Innovation Center, Airfinity, Our World in Data, the World Bank Group, the Asian Development Bank, and the African Vaccine Acquisition Task Team. We focused on accurate data collection. Two authors (N.V. and M.J.) reviewed all the titles, abstracts, and full-text reports independently. Any disagreements between the authors during the study selection were resolved by consensus. The authors independently extracted data on outcomes from all the studies. Data were extracted using a standardized data extraction form. The MeSH terminology used for the search strategy included “COVID-19”, “cancer”, “hesitancy”, “indecision”, “skepticism”, “wavering”, “prevention”, “diagnosis”, “treatment”, “surgery”, “chemotherapy”, and “radiotherapy”. We obtained 116 studies. Among them, there were two guidelines and two reports of taskforces; the rest were all observational studies or questionnaires.

## 3. COVID-19 and Cancer

Patients with cancer and other immunocompromised states were considered susceptible to COVID-19 infection. The diagnosis of COVID-19 in 1% of 1590 patients with cancer was much higher than the incidence of COVID-19 (0.29% per 100,000 people) in the historical control [2]. Patients with cancer were found to have a fatality rate of 5.6% according to COVID-19 outcome data. According to records obtained from the Public Health England National Cancer Registration and Analysis Service, patients with stage II or III bladder, lung, esophagus, ovary, liver, pancreas and stomach cancers with COVID-19 infection experienced a >30% reduction in survival at 6 months and a >17% reduction in survival at 3 months [9]. The Chinese Center for Disease Control and Prevention described the epidemiological characteristics of 72,314 COVID-19 cases in mainland China as of February 11, 2020. They reported that 107 patients (0.5%) had cancer, and 6 of them died. The case fatality was 5.6%, which is higher than the overall reported case fatality (2.3%) from COVID-19 [2]. In a retrospective study of 1253 patients with cancer, the all-cause 30-day mortality was 2.4%, 38.3% and 69.4% in patients with mild, moderate, and severe COVID-19, respectively. COVID-19 was mild in 81% of patients, moderate in 13%, and severe in 6%. The severity of COVID-19 or death was independently associated with increasing age, smoking history, concurrent comorbidities, and palliative intent of treatment [10]. In a study of 128 hospitalized patients with hematological cancers in Wuhan, there was a 10% incidence of COVID-19, significantly higher than the 1% incidence reported for patients with other cancers [11].

During the COVID-19 pandemic, patients with cancer had a higher mortality rate. In a systematic review and meta-analysis of 22 studies that evaluated the prevalence of cancer among patients infected with COVID-19, cancer patients were found to have a higher risk of severe/critical COVID-19 disease, mortality, ICU admission, and mechanical ventilation [12]. A multicenter study of patients with cancer and COVID-19 reported a nearly three-fold increase in the death rate compared to COVID-19 patients without cancer, and patients with cancer also had a much higher severity of illness [13]. There were 9385 deaths reported in New York State in April 2020 from COVID-19. Out of these individuals, cancer patients comprised 8.4% [14]. There was a higher overall case-fatality rate reported in Italy (7.2%) than in China (2.3%) [15]. In another systematic review of 52 studies involving 18,650 patients with COVID-19 and cancer, there were 4243 deaths recorded in this population, with a mortality rate of 25.6% [16].

However, in a study by Mehta et al., out of 3101 cancer patients who were admitted to the hospital, 186 of them tested positive for COVID-19 and the mortality rate was 14.52%. A univariate analysis showed that the risk of death was significantly associated with the presence of comorbidities, especially diabetes and cardiovascular diseases. Anticancer treatments, including chemotherapy, surgery, radiotherapy, targeted therapy, and immunotherapy, administered within a month before the onset of COVID-19 symptoms had no significant effect on mortality [17]. An analysis of 1590 patients studied in 570 Wuhan hospitals showed that COVID-19 infections occurred in 12 patients who had recovered from their previous cancer. It was concluded that COVID-19 is a highly contagious infection for everyone, and cancer survivors are not particularly prone to this infection [2]. Other comorbidities, including age, may skew the incidence of COVID-19 in cancer patients. The median age was 63.1 years in cancer patients versus 48.7 years in subjects without cancer [18]. A study of 890 patients diagnosed with COVID-19 and cancer revealed a mortality rate of 33.6%. This was influenced by the male gender, age 65 or older, and comorbidities. There was no worsening of COVID-19 severity or mortality because of chemotherapy, targeted therapy, or immunotherapy [19]. In a racially diverse group of 27 patients with breast cancer and COVID-19, the majority (74%) did not require hospitalization unless there were multiple co-morbidities [20]. In a prospective study of 800 patients with a diagnosis of cancer and symptomatic COVID-19, there was no significant effect on mortality found for patients with immunotherapy, hormonal therapy, targeted therapy and radiotherapy used within the previous 4 weeks. Mortality from COVID-19 in cancer patients was found to be principally driven by age, gender, and comorbidities [21].

## 4. Cancer, Immune Dysregulation, and Susceptibility to COVID-19 Infection

Cancer patients’ morbidity and mortality increased due to the immunosuppressive state associated with COVID-19 infection, in addition to immune derangement secondary to cancer. Immune escape in cancer is promoted by T-regulatory cells that inhibit transforming growth factor-β (TNF-β) signaling in intra-tumoral T-lymphocytes but not in the lymph nodes draining the tumor. As a result, the cytotoxic T-cell function is impaired [22]. The altered TNF-β signaling also suppresses the innate immune system and reprograms the tumor microenvironment [23]. COVID-19 patients may have poor clinical outcomes due to immune dysregulation and prolonged inflammation. COVID-19 infection alters four genes (ANXA3, GNS, HIST1H1C, RASA3) as well as three genes (HBA1, TFRC, GHITM) related to aging, both of which alter immune responsiveness, increasing susceptibility to infection and cancer [24]. Autopsy reports of patients with COVID-19 revealed low numbers of CD8-positive T-lymphocytes infiltrating the lung tissue as evidence of adaptive immune dysfunction [25]. Clinical reports have also indicated decreased CD4, CD8, and natural killer (NK) cell populations in peripheral blood in COVID-19 infection [26]. Compared to healthy controls, T-cells from patients with COVID-19 express significantly more of the inhibitory programmed death (PD1) molecules, with enhanced immune evasion in patients with cancer [27]. Hence, cancer as well as treatment for cancer, including chemotherapy, made patients more susceptible to COVID-19, resulting in increased mortality and morbidity [28].

## 5. Severity of COVID-19 with Cancer Therapy

Evidence has been mixed regarding COVID-19 severity in cancer patients [29]. According to the OnCovid registry in Europe, mortality rates in patients with cancer diagnosed with COVID-19 have improved over the last 6–8 months [29]. In a study of 1253 patients with cancer who contracted COVID-19, most patients had a mild form of the disease [10]. In a tertiary referral cancer center, out of 1088 tested patients, 186 were positive for COVID-19, although anticancer therapies were not associated with increased mortality [17]. It remains of concern that postponing cancer treatment reduces its therapeutic efficacy and curability rates [30].

## 6. COVID-19 Vaccination Status and Cancer

The vaccination status may affect the COVID-19 disease outcomes and mortality of cancer patients (Figure 2). COVID-19 vaccination was a continuation of the treatment paradigm during the COVID-19 pandemic. Patients with solid tumors with normal cellular immunity were prevented from receiving the COVID-19 vaccine, either because of patients’ reluctance or health-care providers’ hesitancy in giving the vaccine because of the immunocompromised state—further restricting the oncological treatment plan. COVID-19 is more likely to infect those who have undergone radiotherapy, chemotherapy, and immunotherapy, as well as those who have recently undergone bone marrow or stem cell transplant. These patients were considered unable to mount an immune response sufficient enough and so to have a greater chance of getting infected with COVID-19 [31]. There is a difference in response to COVID-19 vaccines among cancer patients. The COVID-19 vaccines are unlikely to induce a humoral response in lymphoma patients treated with an anti-CD20 antibody. The length of time since the last exposure to anti-CD20 antibodies predicts an elevated antibody titer and a more favorable response [32]. Immune response rates were described to be lower amongst cancer patients, especially those with hematologic cancers and those receiving chemotherapy, radiotherapy, or immunosuppressants. Patients with hematologic malignancies, especially lymphoproliferative disorders, have the lowest seroconversion rates. Even after complete immunization, patients with cancer are more likely to get COVID-19 [33]. Patients with multiple myeloma are immunocompromised due to disorders of humoral and cellular immunity as well as the use of immunosuppressive drugs. When compared to healthy controls, the antibody response to the COVID-19 mRNA vaccine appears to be attenuated and delayed after the initial dose in multiple myeloma patients [34]. Nevertheless, a sufficient immune response was still generated in many patients, and vaccination was overall described to be safe and well-tolerated [35]. Mori et al. found no significant difference between the seroconversion rates for AML in remission and MDS compared to healthy controls, with 94.7% and 100%, respectively [36]. COVID-19 is unlikely to be worsened by anti-cancer therapies in cancer patients, including those with hematological malignancies [37].

## 7. COVID-19 Vaccination Hesitancy in Patients with Cancer

In a survey of 2691 people with solid organ tumors, it was found that hesitancy was higher among young, female, non-dominant English speakers and regional residents, as well as among those with non-genitourinary cancers. The vaccine uptake was higher in people who were older, male, metropolitan, spoke English as a first language, and had been diagnosed with cancer for more than six months [38]. In a cross-sectional online study of adults with a history of cancer, among 1073 respondents, 84% of them indicated positive intent toward COVID-19 vaccination, 10% were undecided, and 6% indicated negative attitudes [39]. In a two-wave survey of 2272 individuals between December 2020 and June 2021, among hematologic malignancy patients and survivors, being younger, unmarried, trusting local faith leaders, and not having a bachelor’s degree or more were negatively associated with getting vaccinated. Among those who were hesitant, those with a distrust of vaccines in general were least likely to get vaccinated [40]. Among a total of 424 families with childhood acute lymphoblastic leukemia in remission, 21.4% agreed, 39.6% hesitated, and 38.9% parents disagreed with the vaccination. The most common reason that kept parents from vaccinating their children was a lack of recommendations from professionals [41]. Mistrust of the health-care system, the misconception that COVID-19 vaccination is contraindicated in patients with breast cancer, not having a close acquaintance already vaccinated against COVID-19, noncompliance with prior influenza immunization, being aged younger than 60 years, having low educational attainment and not having a close acquaintance deceased due COVID-19 were all causes of the refusal of COVID-19 vaccines [42]. No study cited concerns about efficacy as a reason for refusal of the vaccine [43] (Table 1).

According to a study of 111 adult cancer patients from a single institution, the main reason for vaccine refusal was incompatibility with patients’ disease or treatment. In this study, 61.3% felt more vulnerable to COVID-19 than the general population. About 55% of the patients were ready to be vaccinated and 14.4% refused the vaccine. The authors concluded that the majority of the patients in this institutional sample accepted the COVID-19 vaccine [44]. In another study of 1001 patients from five institutions, among those who participated in the survey, 293 were cancer patients. In this group, 53.9% were concerned about developing vaccine-related adverse events and 23.5% believed that the vaccine had a negative impact on cancer treatment [45].

## 8. Mitigation Strategies

There was a positive correlation between vaccination status and the trust respondents had for their oncologist, federal agencies, and pharmaceutical companies. It was reported that oncologists and primary care physicians were the most trusted sources of vaccines [40]. The major reasons behind vaccine hesitancy appear to be inadequate information about safety and seeing no reason for the vaccine. It may be possible to circumvent such barriers by providing appropriate information and counseling [46,47,48,49]. To effectively manage cancer patients during the COVID-19 pandemic, it is critical to effectively communicate and educate, providing positive reinforcement during the discussion [50,51,52]. The management strategies for treating cancer patients to avoid COVID-19 infection should include hand hygiene, infection control measures, avoiding high-risk exposure, and educating them about the signs and symptoms of COVID-19 [1]. The main reasons for hesitancy with regard to COVID-19 vaccines, including the rapid pace of vaccine development, safety of the vaccine, questionable long-term effects of the vaccine, misinformation on social media, doubts about the efficacy of the vaccine, general lack of trust in government and pharmaceutical industry, presumed harmless nature of COVID-19, the polarized sociopolitical environment, and the inherent complexities of large-scale vaccination, are all concerns for cancer patients [53]. To achieve a set target in terms of COVID-19 vaccination, each of the above issues must be addressed.

Several factors may complicate efforts to increase vaccine confidence among cancer patients. These factors include the underrepresentation of cancer patients in COVID-19 vaccine trials and specific recommendations concerning vaccine administration and timing for cancer survivors. A potential benefit of vaccine communication efforts targeting survivors is the consideration of social norms, perceptions of risk, and trust. Nevertheless, additional behavioral research is needed in order to better understand and respond to the drivers of vaccine hesitancy among survivors and ensure optimal protection against COVID-19 for this high-risk population [54].

## 9. Management of Cancer and COVID-19

A report by Liang et al. suggests a delay in adjuvant chemotherapy and elective surgery for stable cancer patients with COVID-19. A second recommendation is to strengthen personal protection for cancer patients and cancer survivors. Authors suggest that the COVID-19 infection should be treated more intensively when cancer patients contract this infection, especially if they are older or have other comorbid conditions [2]. Arranging vaccination during the interval between the administration of anticancer drugs, changing the treatment schedule for chemotherapy because of vaccination, avoiding being vaccinated when the white blood cell count is lowered by anticancer drugs, avoiding vaccination on the same day as molecular targeted drugs and avoiding administering anticancer drugs during the week of vaccination (allowing about one week after chemotherapy before vaccination) are some of the suggested strategies for cancer patients [55].

## 10. Limitations of the Study

There are some limitations to this narrative review, including the online search. The search was not comprehensive by any means, and it is possible that some literature has been inadvertently omitted from this review, such as abstracts presented at conferences outside of Europe and America as well as unpublished studies, due to the nature of the search methodology used. During the COVID-19 pandemic, the quality of collected data was impacted by the infrastructure and economic background of each country’s health-care system. There is a concern that the data concerning COVID-19 may not be entirely accurate, as it could be incomplete or have been collected using disparate methods.

## 11. Discussion

To the best of our knowledge, this is the first paper to identify hesitancy as a major cause of health-care deficits for cancer patients during the COVID-19 pandemic. Our review of the data found that there were fewer reported cases of cancer diagnosis, interventions, and palliative care services during the COVID-19 pandemic. Due to the immunocompromised state and the vulnerability to the harmful effects of COVID-19 infection in cancer patients, the hospital posed a constant threat not only to the patient but also to the nursing and medical staff, making it difficult to provide appropriate treatment for cancer patients [2,56].

Adjuvant chemotherapy was delayed significantly because of the fear of acquiring COVID-19 in health-care facilities during the pandemic, leading to reduced survival [57]. It has been difficult to quantify the benefit of intervention for cancer during the pandemic, although reports have confirmed the beneficial effect of radiotherapy and chemotherapy [58]. Patients with cancer were at a higher risk of severe COVID-19, while quarantines, social distancing measures, and interruptions to routine health-care delivery disrupted cancer therapy delivery. The delay in treatment during the COVID-19 pandemic led to poor outcomes, especially in patients with cancer who had spinal cord compression (this is because if immediate decompression is not performed within 24 h, the delay will result in an irreversible neurological deficit) and hypercalcemia, poor evaluation of the lung mass, delayed chemotherapy for testicular and rectal cancers and delayed radiation therapy for gynecological, rectal, lung and head and neck cancers [56].

Health-care organizations have a crucial role to play in eliminating vaccine hesitancy by dispelling disinformation, engaging the public with vaccine education and finding common ground to accept vaccines, while also partnering with state and local governments to boost vaccination rates. These approaches will reduce the risk of contracting COVID-19 [59]. Vaccine enthusiasm increases when positive reinforcement is used in communication. In addition to encouraging family, friends, coworkers, and community members to get vaccinated, the recommendation of the physician is strongly associated with vaccine enthusiasm [60].

COVID-19 infection is now considered a chronic disease. A once-a-year COVID-19 vaccine has been proposed by the Infectious Diseases Society of America and the Food and Drug Administration, which underscores the potential persistence of the virus for years to come. We still do not know how to manage cancer patients when faced with future viral outbreaks. There are no clear guidelines as to how to proceed with screening (breast, colon and prostate), initial therapy (surgical, chemotherapy and radiation therapy) or metastatic cancer management. Some of the strategies for dealing with cancer patients during the COVID-19 pandemic may not be applicable due to recent changes in antiviral therapy and COVID-19 vaccination. However, hesitancy or reluctance to offer either prophylactic or curative therapy and vaccination advice for cancer patients resulting in health-care deficits continues to be a growing concern. The hesitancy of patients to seek medical help during the COVID-19 pandemic is not the sole reason for the health-care failure, as the health-care providers, government agencies, the pharmaceutical industry and administrators must take equal responsibility. Even though there is no simple solution, proactive, compassionate and effective communication with patients that takes into account their cultural diversity and understanding may improve cancer care.

## 12. Conclusions

In conclusion, dealing with hesitancy in relation to cancer care with COVID-19 requires a broader approach with a variety of partners, such as religious, political and community leaders, organizational leaders, health-care providers, and social media such as television, online forums, news media, and patient advocates, all playing an equal role in helping cancer patients with the ever-present fear of contracting COVID-19 infection. Several interventions are essential in the fight against cancer, such as standing orders in the physician’s office, patient reminders, and home health visits. Educating the most vulnerable populations about vaccination is essential to removing the fear of vaccination. Patients with comorbidities, the elderly, obese individuals, and pregnant women are among the groups at risk.

During any future pandemic, cancer patients need timely cancer therapy and appropriate vaccination advice against the virus when it becomes available. Screening patients for the virus prior to chemotherapy infusion and keeping infusion centers open as usual may avoid treatment delays. Patients on parenteral therapy may be switched to an alternative oral chemotherapeutic agent if available. Using home health agencies for home infusion, blood draws for follow-up blood tests, nutritional advice and home physical and occupational therapy may be considered if appropriate. To avoid health-care deficits during the pandemic, health-care personnel must commit to providing continued therapies to cancer patients.

## Figures and Tables

**Figure 1 cancers-15-03115-f001:**
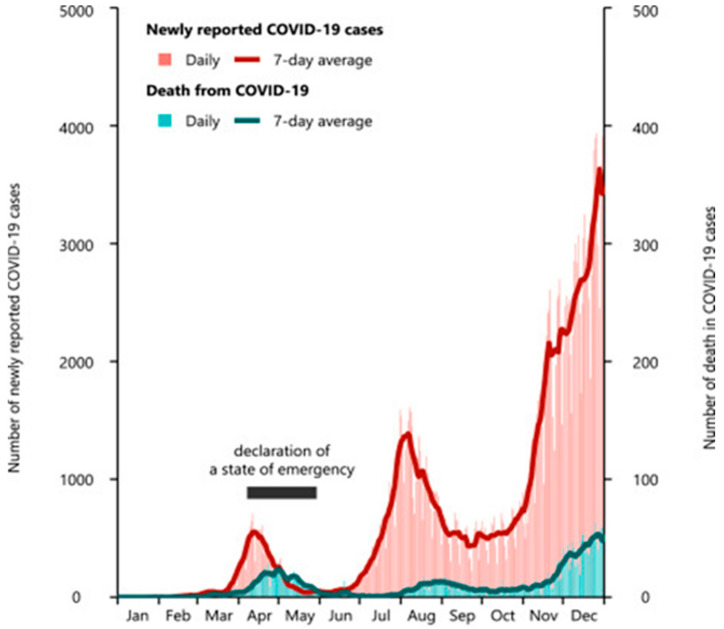
Upsurge in the incidence of COVID-19 in Japan from July–December 2022. This image was derived from WHO data (accessed on 15 March 2023) by Servier, licensed under a Creative Commons Attribution 3.0 Unported License.

**Figure 2 cancers-15-03115-f002:**
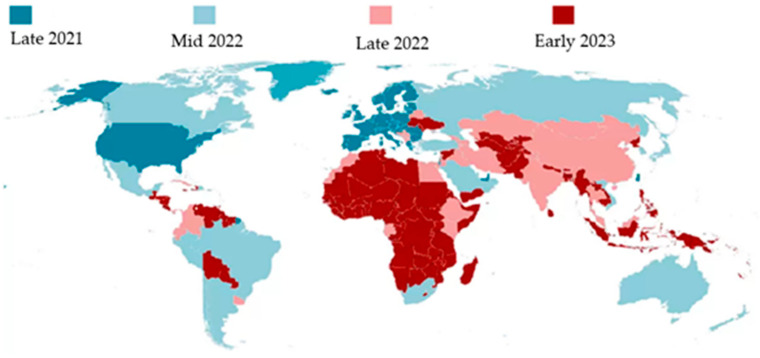
Timeline for global COVID 19 vaccinations—Image modified from the BBC (accessed on March 15, 2023) by Servier, licensed under a Creative Commons Attribution 3.0 Unported License.

**Table 1 cancers-15-03115-t001:** Summary of vaccine hesitancy studies in cancer patients.

Author (year)	Study Design	No. of Participants	Main Findings	Major Concerns among Unvaccinated Population
Bain et al. (2022) [38]	Cross-sectional survey	2691	Higher hesitancy scores were significantly associated with female gender, younger age and non-English speakers	Concerns about adverse effects, interaction with anticancer treatment, increased fear of cancer-related death than COVID-19.
Nyuyen et al. (2022) [39]	Cross-sectional survey	1073	Higher vaccine uptake was significantly associated with older age, male gender, English speakers, long-term cancer diagnosis, and not being on current anticancer treatment. No correlation of hesitancy scores with lower educational level or household income.	Distrust in government/health-care services, risk of thrombosis, desire for more information, fear of interaction with anticancer treatment.
Akesson et al. (2022) [40]	Cross-sectional survey	2272	Higher vaccine hesitancy was associated with female gender, younger age, unmarried, and not having a bachelor’s degree.	Concerns about sufficient vaccine testing, concerns about side effects, concerns about efficacy, survived prior COVID-19 without vaccine.
Villarreal-Garza et al. (2021) [42]	Cross-sectional survey	619	Hesitancy was associated with mistrust of health-care services, not having a close vaccinated family member, and low education level.	COVID-19 vaccines can cause infection, are contraindicated in patients with breast cancer, efficacy concerns, carry a computer chip to surveil the population, could cause infertility.
Haddad et al. (2022) [43]	Cross-sectional survey	240	Hesitancy scoring not performed.	Concerns about side effects, not wanting to be the first, anti-vaxxer stance, government/health-care mistrust, inadequate information.
Moujaess et al. (2022) [44]	Cross-sectional survey	111	Hesitancy scoring not performed.	Interaction with anticancer treatment, increased fear of cancer-related death than COVID-19, government/health-care mistrust, inadequate information, lack of data regarding efficacy among cancer patients.

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
