# Peer review of "Insights into the Impact of Hesitancy on Cancer Care and COVID-19"

_cancers, 2023, doi:10.3390/cancers15123115_

Round 1
Reviewer 1 Report
Dear Authors,
the article deals with an important topic by examining whether there were health care deficits during the COVID-19 pandemic. The question relates primarily to subjective influences that led to non-utilisation. According to the authors, the article is a review.
The following aspects should be considered when reviewing the article:
1. The execution of the article is not sufficiently oriented towards the research question. There are two possible solutions here, so that essential and relevant findings of the article are not lost: The parts of the article that do not fit the research question are assigned to the background. Alternatively, the research question could be expanded.
2. Why was the data in figure 1 selected?
3. In the introduction, it should be explained what is meant by hesitancy and how this is operationalised in the study - the article's double reference to utilisation on the one hand and vaccination on the other should also be addressed here. In the methods section, this should be taken up again and specified for the search carried out. In the results section, too, the concept of hesitancy on which the different studies were based should be precisely described.
4. The methodology as a whole should be revised and sharpened. What form of review is it? I would assume that it is a narrative review. If this assumption is correct, the following aspects should be added: How was the search conducted? How many hits were obtained? What inclusion and exclusion criteria were used.
5. A revision of the results section should be made (see comments at point 1). In addition, it would be helpful if there were an overview of the selected studies so that basic data that are typically considered in systematic reviews are known. This could also be included in the appendix. In the sections, the argumentation could be sharpened; in particular, it must be ensured that it is always possible to clearly classify the results.
6. Are there no results on hesitancy in utilisation? It is unclear which results chapter 9 refers to.
7. Not all results on hesitancy in COVID-19 vaccination refer to patients with cancer. These should be removed from this section (lines 206-206).
8. The discussion should ensure that the conclusions drawn there are also supported by the results presented.
9. The paper should be revised very thoroughly; also the citation needs to be improved.
ok
Author Response
Dear Authors,
The article deals with an important topic by examining whether there were health care deficits during the COVID-19 pandemic. The question relates primarily to subjective influences that led to non-utilisation. According to the authors, the article is a review.
The following aspects should be considered when reviewing the article:
- The execution of the article is not sufficiently oriented towards the research question. There are two possible solutions here, so that essential and relevant findings of the article are not lost: The parts of the article that do not fit the research question are assigned to the background. Alternatively, the research question could be expanded.
Thanks for your comment
As you have mentioned, this article aimed to see whether there were health care deficits during the COVID-19 pandemic in patients with cancer.
The research question has been expanded to ascertain if hesitancy or reluctance played a major role during the pandemic in management of cancer patients.
The following paragraph is added to the introduction:
Healthcare deficits during COVID-19 were largely caused by hesitancy or reluctance during the COVID-19 pandemic. This delayed cancer diagnosis and treatment. During the global pandemic, not only were patients reluctant to travel to the cancer center, but also physicians were reluctant to see cancer patients in the hospital because of constraints. This is either because the clinicians were working remotely or because the healthcare providers were overwhelmed by COVID-19 victims. This resulted in delayed clinic visits, cancellation of preventive measures including mammography, colonoscopy, PSA screening, avoiding endoscopy, ENT/ophthalmological procedures and disrupting surgery, chemotherapy and radiation therapy. During the COVID-19 pandemic, the pharmaceutical industry and other agencies responded by restricting/closing the drug trials either because of financial constraints or lack of personnel to monitor these programs. Finally, when the COVID-19 vaccines were freely available, the hesitancy of cancer patients and providers prevented the vaccination of cancer patients either because of assumed lack of response from the underlying immunocompromised state or lack of response from chemotherapy or fear of the detrimental effect of the vaccine (venous and arterial thromboembolism). This further restricted cancer patients' access to life-saving therapy. This review will explore how delays in seeking medical care and reluctance to receive COVID-19 vaccinations have affected cancer patients.
- Why was the data in figure 1 selected?
Thanks for your comment
Just when the COVID-19 pandemic was thought to be under control, developed country like Japan despite its well-organized national vaccination program reported a cumulative total of 16,423,053 infected and 36,234 deaths as of August 19, 2022. This was mainly from the Omicron variant of COVID-19 which led to the 7th wave of the pandemic and statewide restriction of movement of citizens in Japan. (Ref: COVID-19 in Japan during 2020-2022: Characteristics, responses, and implications for the health care system K Karako, P Song, Y Chen, T Karako - Journal of Global Health, 2022). Following this, COVID-19 booster targeting Omicron and other variants was introduced
(Ref: COVID-19 in Japan during 2020-2022: Characteristics, responses, and implications for the health care system K Karako, P Song, Y Chen, T Karako - Journal of Global Health, 2022). Following this, COVID-19 booster targeting Omicron and other variants was introduced
- In the introduction, it should be explained what is meant by hesitancy and how this is operationalised in the study - the article's double reference to utilisation on the one hand and vaccination on the other should also be addressed here. In the methods section, this should be taken up again and specified for the search carried out. In the results section, too, the concept of hesitancy on which the different studies were based should be precisely described.
Thanks for your comment
See the response to point 1 regarding hesitancy
We have added to the methodology section the inclusion and exclusion criteria and the criteria for selection of the studies for this narrative review. We will add a table in the results section to include the studies in the text
- The methodology as a whole should be revised and sharpened. What form of review is it? I would assume that it is a narrative review. If this assumption is correct, the following aspects should be added: How was the search conducted? How many hits were obtained? What inclusion and exclusion criteria were used.
Thanks for your comment
See above.
- A revision of the results section should be made (see comments at point 1). In addition, it would be helpful if there were an overview of the selected studies so that basic data that are typically considered in systematic reviews are suboptimal response to vaccination because of age of use of rituximab and other monoclonal antibodies or inherent defect of the immune system to mount adequate response to the vaccine)
Thanks for your comment
We have constructed a table giving an overview of the selected studies. We have discussed the suboptimal response to vaccination because of use of rituximab and other monoclonal antibodies or inherent defect of the immune system to mount adequate response to the vaccine in relevant sections
- Are there no results on hesitancy in utilisation? It is unclear which results chapter 9 refers to.
Thanks for your comment
During the peak of the pandemic patients with cancer were sidelined because of the more pressing issues in managing COVID-19 infected patients. Healthcare personnel worked offsite in most healthcare facilities in the country. Even though telehealth helped to a certain extent, lack of clinical examination did not help cancer patients. This was never cited in published literature as the reason for healthcare deficits during the pandemic. In chapter 9, the authors provide their views regarding continuing chemo/radiotherapy but these are not standard guidelines. Cancer care during COVID-19 was evolving and even the local protocols and guidelines during the pandemic for managing these patients become obsolete with specific COVID-19 therapy and the vaccination program. Response to COVID-19 vaccination for cancer patients was suboptimal but it is now known that patients with solid tumors and acute myeloblastic leukemia during remission have the same response to the vaccine as the normal population.
- Not all results on hesitancy in COVID-19 vaccination refer to patients with cancer. These should be removed from this section (lines 206-206).
Thanks for your comment
Yes, we agree. Initially we thought some of these factors were applicable to normal population, but we have decided to delete this because of lack of published data in cancer patients
- The discussion should ensure that the conclusions drawn there are also supported by the results presented.
Thanks for your comment
We have added the following paragraphs to the discussion and conclusion sections
" The COVID-19 infection is now considered a chronic disease. A once-a-year COVID-19 vaccine has been proposed by the Infectious Diseases Society of America and the Food and Drug Administration, which underscores the potential persistence of the virus for years to come. We still do not know how to manage cancer patients when faced with future viral outbreaks. There are no clear guidelines as to how to proceed with screening (breast, colon and prostate), initial therapy (surgical, chemotherapy and radiation therapy) or metastatic cancer management. Some of the strategies in dealing with cancer patients during the COVID-19 pandemic may not be applicable with the recent changes to antiviral therapy and Covid 19 vaccination. However, the hesitancy or lack of willingness to offer either prophylactic or curative cancer therapy resulting in healthcare deficits continues to be a growing concern. The hesitancy and reluctance of patients to seek medical help during the COVID-19 pandemic is not the sole reason for the healthcare deficit. However, the healthcare providers, government agencies, the pharmaceutical industry and administrators should take equal responsibility. Even though there is no simple solution to this, being proactive, compassionate and effective communication taking into account the cultural diversity and understanding may improve cancer care.
In conclusion section,
" The treatment of cancer patients with ongoing viral infections does not have a specific solution. Getting vaccinated, when the vaccine is available is the only way to prevent the viral infection and proceed with ongoing cancer therapy. It is important to streamline the management of outpatient appointments, scheduled treatments, and hospitalizations for cancer patients in the future. As solid tumor patients are not considered to have a higher risk of COVID-19, the indicated cancer therapy should be offered. Screening cancer patients for COVID 19 prior to chemotherapy infusion and maintaining chemotherapy infusion centers open as usual may avoid treatment delays. Patients on parenteral therapy may be switched to an alternative oral chemotherapeutic agent if appropriate. Home infusion services can be considered if applicable. To avoid exposure, blood tests can be done at home, by home health agencies. To avoid healthcare deficits during the pandemic, healthcare personnel must commit to providing continued therapies to cancer patients.
- The paper should be revised very thoroughly; also the citation needs to be improved.
Thanks for your comment
Yes, the paper has been revised by the authors
Reviewer 2 Report
This manuscript by Visweshwar et al., represents a nice overview of the impact of COVID- 19 infection on cancer outcomes, cancer treatment pathway, as well as the attitudes of cancer patients towards vaccination.
Questions/ comments to the authors:
Line 34- The authors mention that one of the aspects of this review is to present impact of COVID-19 on entry to clinical trials. My impression is that this particular aspect is not adequately addressed in the manuscript.
Line 204- what does the percentage 21.4% refer to?
Lines 299-300 need more clarification
The manuscript requires minor editing of the language to enhance understanding and clarity of what authors wish to convey
Author Response
Comments and Suggestions for Authors
This manuscript by Visweshwar et al., represents a nice overview of the impact of COVID- 19 infection on cancer outcomes, cancer treatment pathway, as well as the attitudes of cancer patients towards vaccination.
Questions/ comments to the authors:
Line 34- The authors mention that one of the aspects of this review is to present impact of COVID-19 on entry to clinical trials. My impression is that this particular aspect is not adequately addressed in the manuscript.
Thanks for your comment
The hesitancy or reluctance played a major role during the COVID-19 pandemic in the diagnosis and treatment of cancer patients - the lack of interest of the pharmaceutical industry and other agencies restricting/closing their drug trials because of the financial restrictions or lack of personnel to monitor these programs was only part of cancer patients' failure to receive appropriate therapy. The other reasons for lack of effective cancer treatment during COVID-19 pandemic was because of the inability of patients to avoid travel to the cancer center because of the travel restrictions from "lock down" during the global pandemic but also lack of physicians to see cancer patients in the health care facility. This is either because clinicians worked remotely or because health care workers were overwhelmed by COVID-19 victims. This delayed cancer diagnoses and treatment resulting in lack of clinic visits, cancelled preventive measures including mammography, colonoscopy, PSA screening, scheduling endoscopy, ENT/ophthalmology procedures and disrupted surgical, chemotherapy and radiation therapy. Finally, when the vaccines became freely available, the hesitancy of cancer patients and providers restricting the vaccination to cancer patients because of either assumed lack of response from the underlying immunocompromised state, or presumed lack of response from the chemotherapy or because of the detrimental effect of the vaccine (venous and arterial thromboembolism) further restricted access to life-saving therapy
Line 204- what does the percentage 21.4% refer to?
Thanks for your comment
A total of 424 families with childhood acute lymphoblastic leukemia survivors were included in the study. Among them, 91 (21.4%) agreed, 168 (39.6%) hesitated, and 165 (38.9%) parents disagreed with the vaccination (Ref: Parent Acceptance toward Inactivated COVID-19 Vaccination in Children with Acute Lymphoblastic Leukemia: The Power of Oncologist and Alliance - Ma et. al, Vaccines 2022, 10(12), 2016).
Lines 299-300 need more clarification
Thanks for your comment
The delay in treatment during COVID-19 pandemic had poor outcome especially in patients with cancer having spinal cord compression - this is because if the immediate decompression is not performed within 24 hours, the delay will result in irreversible neurological deficit
Comments on the Quality of English Language
The manuscript requires minor editing of the language to enhance understanding and clarity of what authors wish to convey
Reviewer 3 Report
(Attached Word doc; also pasted below)
Overall comments
This is an excellent contribution to the literature as it compels readers to consider the compounding complications of cancer treatment during health crises.
Minor suggestions
1. This may be for the typesetting stage, but please be aware of the inconsistent font.
2. Consider a more compelling title that matches your key theme. Example: "Pandemic Paralysis: The Impact of COVID-19 on Cancer Care and Patient Hesitancy"
3. If it’s in accordance with the journal norms, consider callout boxes of key points, findings, and compelling messages.
a. E.g., Key theme: The barriers and challenges faced by cancer patients and healthcare providers during the COVID-19 pandemic, including the effects of delayed diagnoses, disrupted treatments, and vaccination hesitancy on cancer care outcomes.
Suggestions
1. Figure 1 is an odd choice to reflect the early stages of the global pandemic. (And I wonder if the caption gives the wrong year – 2022 looks more like 2020) Suggestion: choose a global depiction of the same or describe why Japan is a good representation of the outbreak profile. (This is reasonable if you make the connection that most developed countries followed this amplified outbreak cycle, with each recurrence being much larger than the last throughout 2020).
2. On line 19, please amend the phrase “During the COVID-19 pandemic, cancer patients were most affected” to include “were among the most affected.” The most affected were elderly patients and those with severe comorbidities (including cancer). (Of course, these compound at the intersection, e.g., risk increases for those who are elderly AND have comorbidities AND have cancer).
3. Please consider revising the Simple Summary (lines 19-27). Some example sentences:
The impact of COVID-19 on cancer diagnosis and management during the pandemic remains uncertain. The risk of infection and travel restrictions made some cancer patients reluctant or unable to travel, hindering their access to diagnostic procedures and treatment. The pandemic also led to a shortage of healthcare personnel, forcing cancer centers to limit or postpone screening procedures such as mammography and colonoscopy, as well as elective surgeries, chemotherapy, and radiation therapy. Ophthalmic and ENT procedures, endoscopy, intubation, surgery, and chemotherapy infusions were also canceled to minimize close contact during the outbreak. This review will explore how delays in seeking medical care and reluctance to receive COVID-19 vaccinations have affected cancer patients.
4. Please revise lines 32-33, where it currently reads: “Cancer patients were considered to have a higher risk of contracting COVID-19 because of comorbidities and aging.”
Cancer patients faced a higher risk of severe illness and complications if they did contract the virus. You might mention that this increased risk was due to factors such as compromised immune systems from cancer treatments and the presence of comorbidities.
5. Reviewer note: reading citation [1][1] and your introduction, I see that there was a twofold increased risk of contracting SARS-CoV-2 compared to the general population. So please consider amending “Cancer patients have a higher risk of contracting COVID-19 because of comorbidities and aging” to something like
“In addition to being at an increased risk of contracting COVID-19, cancer patients have a higher risk of severe illness and complications if they did contract the virus, compared to the general population.”
6. Please consider revising the opening sentences in the introduction (lines 40-45) to something like:
Currently, some geographic regions, particularly in the Asia-Pacific area, are experiencing the seventh wave of COVID-19, which was declared a global pandemic by the World Health Organization in March 2020 (figure 1). The pandemic has significantly impacted cancer patients. They face an estimated twofold increase in the risk of contracting COVID-19 compared to the general population. Furthermore, they have a 3.5-fold higher risk of developing severe COVID-19, leading to increased morbidity and mortality. These heightened risks are compounded by factors such as aging and the presence of comorbidities among cancer patients.
7. Please consider revising lines 46-48 to reflect viral infections' recurrent and ever-evolving nature. Example:
As the original strain of the virus has receded, the recurrent and ever-evolving nature of viral infections has led to the emergence of virulent strains such as the alpha, delta, and omicron variants. These new strains have been infecting individuals, including those who have already been vaccinated.
8. Please consider revising lines 55-62 to reflect the complexity of the healthcare landscape (especially in 2020). Example:
In the early months and years of the COVID-19 pandemic, the complex healthcare landscape contributed to fewer cancer diagnoses as initial investigations of symptoms that could confirm cancer were delayed due to fear of the virus [6]. Screening programs for breast, colorectal, and cervical cancers were put on hold or suspended as the healthcare system faced unprecedented demands in managing the pandemic [7]. Both patients and healthcare providers, wary of COVID-19, postponed appointments at cancer centers, leading to missed diagnoses and treatment opportunities for cancer patients. This review will examine how hesitancy to seek medical attention and receive COVID-19 vaccinations impacts cancer outcomes.
9. Consider modifying the statement on lines 68-73 to reflect disparate and incomplete data capture methods. Example:
Numerous international agencies, including the World Health Organization, International Monetary Fund, COVAX Global Vaccine Assessment, United Nations Children's Fund (UNICEF), Duke Global Health Innovation Center, Airfinity, Our World in Data, the World Bank Group, the Asian Development Bank, and African Vaccine Acquisition Task Team, have all produced information related to COVID-19. However, there is concern that the data may not be entirely accurate, as it could be incomplete or collected using disparate methods.
10. Figure 2 has an incomplete caption to stand alone. Consider adding text to help with interpretation – also, please update the text around lines 52-53. The text and caption should make it easy for readers to interpret Figure 2. I don’t understand what the figure depicts (color-coded to years to achieve (?)). Are the years color-coded representing years that region achieved herd immunity?
11. Figure 3 needs a caption and connection to a public health model – it does not look like any standard model, e.g., health belief or social cognitive model.
I’d recommend removing Figure 3, as it does not add enough to the discussion.
13. Consider revising your last statement in the Conclusion section (lines 320-321) to reflect the focus on at-risk populations (rather than staving off a future pandemic). Example:
It is crucial to prioritize the needs of at-risk patients, such as those with cancer, during a pandemic or healthcare crisis to minimize the negative impact on their treatment outcomes and overall well-being.
14. Optional: Consider mentioning Champion approaches or positive deviance analysis. Are there any findings from healthcare centers that did phenomenally well (e.g., with vaccination rates among at-risk patients) during the outbreak?
[1] H.O. Al‐Shamsi, W. Alhazzani, A. Alhuraiji, E.A. Coomes, R.F. Chemaly, M. Almuhanna, R.A. Wolff, N.K. Ibrahim, M.L. Chua, S.J. Hotte, A practical approach to the management of cancer patients during the novel coronavirus disease 2019 (COVID-19) pandemic: an international collaborative group, The oncologist 25(6) (2020) e936-e945
I recommend using a grammar application, such as Grammarly, to aid in sentence structure. Overall, the quality was good, but some sentences didn't, for example, have matching tenses.
Author Response
Minor suggestions
- This may be for the typesetting stage, but please be aware of the inconsistent font
Thanks for your comment -the font is made consistent.
- Consider a more compelling title that matches your key theme. Example: "Pandemic Paralysis: The Impact of COVID-19 on Cancer Care and Patient Hesitancy"
Thanks for your suggestion
We have changed the title to. "Insights into the Impact of Hesitancy on Cancer Care and COVID-19 "
- If it's in accordance with the journal norms, consider callout boxes of key points, findings, and compelling messages.
Thanks for your comment
There are no callout boxes of key points in this journal
- E.g., Key theme: The barriers and challenges faced by cancer patients and healthcare providers during the COVID-19 pandemic, including the effects of delayed diagnoses, disrupted treatments, and vaccination hesitancy on cancer care outcomes.
Suggestions
- Figure 1 is an odd choice to reflect the early stages of the global pandemic. (And I wonder if the caption gives the wrong year - 2022 looks more like 2020) Suggestion: choose a global depiction of the same or describe why Japan is a good representation of the outbreak profile. (This is reasonable if you make the connection that most developed countries followed this amplified outbreak cycle, with each recurrence being much larger than the last throughout 2020).
Thanks for your comment
Just when the COVID-19 pandemic was thought to be under control, a developed country like Japan despite its well-organized national vaccination program reported a cumulative total of 16,423,053 infected and 36,234 deaths as of August 19, 2022. This was mainly from the Omicron variant of COVID-19 which led to the 7th wave of the pandemic and statewide restriction of movement of citizens in Japan
(Ref: COVID-19 in Japan during 2020-2022: Characteristics, responses, and implications for the health care system K Karako, P Song, Y Chen, T Karako - Journal of Global Health, 2022). The other reason why we included this statement is because clear statistics of Omicron outbreak is not available from rest of the developed countries
- On line 19, please amend the phrase "During the COVID-19 pandemic, cancer patients were most affected" to include "were among the most affected." The most affected were elderly patients and those with severe comorbidities (including cancer). (Of course, these compound at the intersection, e.g., risk increases for those who are elderly AND have comorbidities AND have cancer).
Thanks for your comment
Appropriate amendment was made
- Please consider revising the Simple Summary (lines 19-27). Some example sentences:
The impact of COVID-19 on cancer diagnosis and management during the pandemic remains uncertain. The risk of infection and travel restrictions made some cancer patients reluctant or unable to travel, hindering their access to diagnostic procedures and treatment. The pandemic also led to a shortage of healthcare personnel, forcing cancer centers to limit or postpone screening procedures such as mammography and colonoscopy, as well as elective surgeries, chemotherapy, and radiation therapy. Ophthalmic and ENT procedures, endoscopy, intubation, surgery, and chemotherapy infusions were also canceled to minimize close contact during the outbreak. This review will explore how delays in seeking medical care and reluctance to receive COVID-19 vaccinations have affected cancer patients.
Thanks for your suggestion
The text was amended accordingly
- Please revise lines 32-33, where it currently reads: "Cancer patients were considered to have a higher risk of contracting COVID-19 because of comorbidities and aging."
Cancer patients faced a higher risk of severe illness and complications if they did contract the virus. You might mention that this increased risk was due to factors such as compromised immune systems from cancer treatments and the presence of comorbidities.
Thanks for your suggestion
The text was amended as follows:
Those with cancer were more likely to suffer severe illness and complications if they contracted COVID-19. A compromised immune system and comorbidities in cancer patients may have contributed to this increased risk.
- 5. Reviewer note: reading citation [1][1] and your introduction, I see that there was a twofold increased risk of contracting SARS-CoV-2 compared to the general population. So please consider amending "Cancer patients have a higher risk of contracting COVID-19 because of comorbidities and aging" to something like
"In addition to being at an increased risk of contracting COVID-19, cancer patients have a higher risk of severe illness and complications if they did contract the virus, compared to the general population."
Thanks for your input
We have amended the test accordingly
- Please consider revising the opening sentences in the introduction (lines 40-45) to something like:
Currently, some geographic regions, particularly in the Asia-Pacific area, are experiencing the seventh wave of COVID-19, which was declared a global pandemic by the World Health Organization in March 2020 (figure 1). The pandemic has significantly impacted cancer patients. They face an estimated twofold increase in the risk of contracting COVID-19 compared to the general population. Furthermore, they have a 3.5-fold higher risk of developing severe COVID-19, leading to increased morbidity and mortality. These heightened risks are compounded by factors such as aging and the presence of comorbidities among cancer patients.
Thanks for your suggestion
We have incorporated this in the text omitting the term “currently”, as this spike happened between July - September 2022
- Please consider revising lines 46-48 to reflect viral infections' recurrent and ever-evolving nature. Example:
As the original strain of the virus has receded, the recurrent and ever-evolving nature of viral infections has led to the emergence of virulent strains such as the alpha, delta, and omicron variants. These new strains have been infecting individuals, including those who have already been vaccinated.
Thanks for your suggestion
We have amended the test as the revised wording is more appropriate
- Please consider revising lines 55-62 to reflect the complexity of the healthcare landscape (especially in 2020). Example:
In the early months and years of the COVID-19 pandemic, the complex healthcare landscape contributed to fewer cancer diagnoses as initial investigations of symptoms that could confirm cancer were delayed due to fear of the virus [6]. Screening programs for breast, colorectal, and cervical cancers were put on hold or suspended as the healthcare system faced unprecedented demands in managing the pandemic [7]. Both patients and healthcare providers, wary of COVID-19, postponed appointments at cancer centers, leading to missed diagnoses and treatment opportunities for cancer patients. This review will examine how hesitancy to seek medical attention and receive COVID-19 vaccinations impacts cancer outcomes.
Thanks for your input
The above changes in the text were made
- Consider modifying the statement on lines 68-73 to reflect disparate and incomplete data capture methods. Example:
Numerous international agencies, including the World Health Organization, International Monetary Fund, COVAX Global Vaccine Assessment, United Nations Children's Fund (UNICEF), Duke Global Health Innovation Center, Airfinity, Our World in Data, the World Bank Group, the Asian Development Bank, and African Vaccine Acquisition Task Team, have all produced information related to COVID-19. However, there is concern that the data may not be entirely accurate, as it could be incomplete or collected using disparate methods.
Thanks for your suggestion
We made appropriate changes in the text
- Figure 2 has an incomplete caption to stand alone. Consider adding text to help with interpretation - also, please update the text around lines 52-53. The text and caption should make it easy for readers to interpret Figure 2. I don't understand what the figure depicts (color-coded to years to achieve (?)). Are the years color-coded representing years that region achieved herd immunity?
Thanks for your comment
Fig. 2 was added to "COVID-19 vaccination status and cancer"- A once-a-year COVID-19 vaccine has been proposed by the Infectious Diseases Society of America and the Food and Drug Administration, which underscores the potential persistence of the virus for years to come. We thought it is important to keep this figure as majority of the global population is still not vaccinated
- Figure 3 needs a caption and connection to a public health model - it does not look like any standard model, e.g., health belief or social cognitive model.
I'd recommend removing Figure 3, as it does not add enough to the discussion.
Thanks for your advice
The authors also agreed that Fig. 3 should be removed
- Consider revising your last statement in the Conclusion section (lines 320-321) to reflect the focus on at-risk populations (rather than staving off a future pandemic). Example:
It is crucial to prioritize the needs of at-risk patients, such as those with cancer, during a pandemic or healthcare crisis to minimize the negative impact on their treatment outcomes and overall well-being.
Thanks for your suggestion
Above sentence was added
We have added the following paragraph in the conclusion section
" There is no specific solution available to take care of cancer patients with an ongoing viral infection. Getting vaccinated, when the vaccine is available is the only way to prevent the viral infection and to proceed with ongoing cancer therapy. In the future, optimal management of outpatient appointments, scheduled treatments and hospitalizations for cancer patients should be streamlined. As solid tumor patients are not considered to have a higher risk of contracting COVID-19, the specific cancer therapy should be offered. Screening of cancer patients for Covid 19 prior to chemotherapy infusion and maintaining chemotherapy infusion centers open as usual may avoid delay in treatment schedule. Patients on active therapy may be switched to an alternative oral chemotherapeutic agent if available. Home infusion services can be considered if applicable. To avoid exposure, blood tests can be done at home by the home health agencies. Above all, commitment to provide continued therapy for cancer patients by the healthcare personnel is needed to avoid healthcare deficits during the pandemic. A once-a-year COVID-19 vaccine has been proposed by the Infectious Diseases Society of America and the Food and Drug Administration, which underscores the potential persistence of the virus for years to come. It is crucial to prioritize the needs of at-risk patients, such as those with cancer, during a pandemic or healthcare crisis to minimize the negative impact on their treatment outcomes and overall well-being”.
- Optional: Consider mentioning Champion approaches or positive deviance analysis. Are there any findings from healthcare centers that did phenomenally well (e.g., with vaccination rates among at-risk patients) during the outbreak?
We are not aware of any particular healthcare facility that did phenomenally well during the pandemic-especially during the Omicron outbreak which was very contagious
[1] H.O. Al?Shamsi, W. Alhazzani, A. Alhuraiji, E.A. Coomes, R.F. Chemaly, M. Almuhanna, R.A. Wolff, N.K. Ibrahim, M.L. Chua, S.J. Hotte, A practical approach to the management of cancer patients during the novel coronavirus disease 2019 (COVID-19) pandemic: an international collaborative group, The oncologist 25(6) (2020) e936-e945
[1] H.O. Al‐Shamsi, W. Alhazzani, A. Alhuraiji, E.A. Coomes, R.F. Chemaly, M. Almuhanna, R.A. Wolff, N.K. Ibrahim, M.L. Chua, S.J. Hotte, A practical approach to the management of cancer patients during the novel coronavirus disease 2019 (COVID-19) pandemic: an international collaborative group, The oncologist 25(6) (2020) e936-e945
Comments on the Quality of English Language
Reviewer 4 Report
Introduction
· Because the landscape of COVID-19 continues to change rapidly, please put a date to anchor “currently” in the first line. This may be outdated at the time of this review (April 2023). My quick search indicates that the “seventh wave” was in summer-early fall 2022. Perhaps just remove that sentence.
· Please use COVID-19 consistently throughout manuscript (rather than Covid-19)
· It is unclear to me what gap in the literature this paper fills
· The purpose statement could be strengthened. It was not completely clear to me what the goal of the paper was.
· One point made in the purpose was related to hesitancy to enter into clinical trials but that is not discussed at all in the paper
Figure 1
· What year is this data? It appears to be 2020 based on the declaration of a state of emergency, but the figure title lists winter 2022.
Data collection
· Insufficient details about how data collection was done. What constitutes accurate data collection? What elements were used? What terms were searched in Medline/Embase?
COVID-19 and Cancer
· What is the reference to support the fatality rate (COVID-19 outcome data)?
· In the sentence “According to the records obtained from Public Health England National Cancer Registration Service for aggressive cancers, patients with stage II or III bladder, lung, esophagus, ovary, liver, pancreas and stomach cancers with COVID-19 infection experienced >30% reduction in survival at 6 months and >17% reduction in survival at 3 months” what is the reduction relative to? To those with one of those cancers without COVID-19?
· “During the COVID-19 pandemic, patients with cancer had a higher mortality rate…” than whom? In general, patients with cancer have a higher mortality rate than patients without cancer.
Severity of COVID-19 with cancer therapy
· Again, please indicate the specific time period indicated by “the last 6-8 months”
Discussion
· Please discuss potential solutions or strategies for addressing the issues raised, beyond broad suggestions for partnership and education
· Please acknowledge any potential limitations or biases in the data or sources used for the review
Overall impressions
This paper does not come together in a cohesive way for me and it is unclear what the main point is. I initially thought that the hesitancy to be discussed was going to be about seeking cancer care out of fear of contracting COVID-19. And thus, poor outcomes are expected due to delayed treatment. But then the paper discusses hesitancy for receiving the vaccine. Perhaps this paper is trying to do too much. Or it needs some work to make it fit together better.
Minor grammatical errors and inconsistency of formatting (i.e. "COVID-19"), but overall no issues with English language
Author Response
Comments and Suggestions for Authors
Introduction
Because the landscape of COVID-19 continues to change rapidly, please put a date to anchor "currently" in the first line. This may be outdated at the time of this review (April 2023). My quick search indicates that the "seventh wave" was in summer-early fall 2022. Perhaps just remove that sentence.
Thanks for your comment
We have altered the date accordingly. Just when the COVID-19 pandemic was thought to be under control, a developed country like Japan despite its well-organized national vaccination program, during 7th wave (July-December 2022) reported a cumulative total of 16,423,053 infected and 36,234 deaths as of August 19, 2022. This was mainly from the Omicron variant of COVID-19 which led to statewide restriction of movement of citizens in Japan (Ref: COVID-19 in Japan during 2020-2022: Characteristics, responses, and implications for the health care system K Karako, P Song, Y Chen, T Karako - Journal of Global Health, 2022). Following this, COVID-19 booster targeting Omicron and other variants was introduced
Please use COVID-19 consistently throughout manuscript (rather than Covid-19)
Thanks for your comment
COVID-19 is used consistently throughout manuscript
It is unclear to me what gap in the literature this paper fills
To our knowledge, this is the first paper examining the underlying causes of healthcare deficits in cancer patients during the COVID-19 pandemic. This paper analyzes the delay in diagnosis and treatment of cancer due to patients' hesitancy to travel to cancer centers and physicians' reluctance to see cancer patients in hospitals. During the peak of the pandemic patients with cancer were sidelined because of the more pressing issues in managing COVID-19 infected patients. Healthcare personnel worked offsite in most healthcare facilities in the country. Even though telehealth helped to a certain extent, lack of clinical examination did not help cancer patients. This was never cited as the reason for healthcare deficits during the pandemic. Healthcare deficits during COVID-19 were largely caused by hesitancy or reluctance. This has delayed cancer diagnosis and treatment. The hesitancy was not only from patients avoiding travel to the cancer center because of the travel restrictions from "lock down" during the global pandemic but also from hesitancy of the physicians seeing cancer patients in the healthcare facility. This resulted in delayed clinic visits, cancellation of preventive measures including mammography, colonoscopy, PSA screening, avoiding endoscopy, ENT/ophthalmological procedures and disrupting surgery, chemotherapy, and radiation therapy. During the COVID-19 pandemic, the pharmaceutical industry and other agencies responded by restricting/closing the drug trials either because of financial constraints or lack of personnel to monitor these programs. Furthermore, when COVID-19 vaccines were freely available, providers were reluctant to administer vaccinations (which were a continuation of preventive treatment paradigms during the COVID-19 pandemic) to cancer patients for a variety of reasons, including presumed lack of response from chemotherapy or because of their immunocompromised state. Fear of the detrimental effect of the vaccine (venous and arterial thromboembolism) also added to the problem.
This further restricted cancer patients' access to life-saving therapy. This review will explore how delays in seeking medical care and reluctance to receive COVID-19 vaccinations have affected cancer patients.
The purpose statement could be strengthened. It was not completely clear to me what the goal of the paper was.
Thanks for your comment
Please see the response to the previous comment
One point made in the purpose was related to hesitancy to enter into clinical trials but that is not discussed at all in the paper
Thanks for your comment
Please see above.
Figure 1
What year is this data? It appears to be 2020 based on the declaration of a state of emergency, but the figure title lists winter 2022.
Thanks for your comment
Please see the previous response to this question
Data collection
- Insufficient details about how data collection was done. What constitutes accurate data collection? What elements were used? What terms were searched in Medline/Embase?
Thanks for your comment
We have added the following paragraph to the on-data collection section
Using databases including Medline and Embase, guidelines from the American Society of Clinical Oncology and the European Society of Medical Oncology, we reviewed the literature for the impact of Covid-19 on cancer from January 2020 through December 2022. We also searched World Health Organization, International Monetary Fund, COVAX Global Vaccine Assessment, the United Nations Children's Fund (UNICEF), Duke Global Health Innovation Center, Airfinity, Our World in Data, the World Bank Group, the Asian Development Bank, and African Vaccine Acquisition Task Team. We focused on accurate data collection. Two authors (NV and MJ) reviewed all titles, abstracts, and full-text reports independently. Any disagreements between authors during the study selection were resolved by consensus. The authors independently extracted data on outcomes from all studies. Data were extracted using a standardized data extraction form. The MeSH terminology used for search strategy included: “COVID-19”; “Cancer”; “Hesitancy”; “indecision”; “skepticism”; “wavering”; “prevention”; “diagnosis”; “treatment”; “Surgery”; “Chemotherapy”; “Radiotherapy”. We obtained 116 studies, out of which there were two guidelines and two reports of taskforces, the rest were all observational studies or questionnaires.
- What is the reference to support the fatality rate (COVID-19 outcome data)?
- In the sentence “According to the records obtained from Public Health England National Cancer Registration Service for aggressive cancers, patients with stage II or III bladder, lung, esophagus, ovary, liver, pancreas and stomach cancers with COVID-19 infection experienced >30% reduction in survival at 6 months and >17% reduction in survival at 3 months” what is the reduction relative to? To those with one of those cancers without COVID-19?
Thanks for your comment
This was an observational data by the authors (A. Sud, M.E. Jones, J. Broggio, C. Loveday, B. Torr, A. Garrett, D.L. Nicol, S. Jhanji, S.A. Boyce, F. Gronthoud, Collateral damage: the impact on outcomes from cancer surgery of the COVID-19 pandemic, Annals of Oncology 31(8) (2020) 1065-1074.)
“During the COVID-19 pandemic, patients with cancer had a higher mortality rate…” than whom? In general, patients with cancer have a higher mortality rate than patients without cancer.
Thanks for your comment
We found a reference for cancer patients during COVID-19 pandemic with cancer have a higher mortality rate than patients without cancer has been added to the text. “The Chinese Center for Disease Control and Prevention described the epidemiological characteristics of 72,314 COVID-19 cases in mainland China as of February 11, 2020. They reported that 107 patients (0.5%) had cancer, and 6 of them died. The case fatality was 5.6%, which is higher than the overall reported case fatality (2.3%) from COVID-19” (Ref:Liang W, Guan W, Chen R et al. Cancer patients in SARS-CoV-2 infection: A nationwide analysis in China. Lancet Oncol 2020;21:335–337).
Severity of COVID-19 with cancer therapy
- Again, please indicate the specific time period indicated by “the last 6-8 months”
Thanks for your comment
We have included the specific period in the text
Discussion
Please discuss potential solutions or strategies for addressing the issues raised, beyond broad suggestions for partnership and education
Thanks for your comment
We have added the following paragraph to the conclusion section
There is no specific solution available to treat cancer patients with ongoing viral pandemic. Getting vaccinated during a viral pandemic when the vaccine is available is the only way to prevent the viral infection and proceed with ongoing cancer therapy. In the future, optimal management of outpatient appointments, scheduled treatments and hospitalizations for cancer patients should be streamlined. As solid tumor patients are not considered to have a higher risk of COVID-19, the specific cancer therapy should be offered. Screening cancer patients for COVID-19 prior to chemotherapy infusion and maintaining chemotherapy infusion centers open as usual may avoid treatment delays. Patients on parenteral therapy may be switched to an alternative oral chemotherapeutic agent if available. Home infusion services can be considered if applicable. To avoid exposure, blood tests can be done at home by health agencies. To avoid healthcare deficits during the pandemic, health care professionals must commit to continued cancer treatment. A once-a-year COVID-19 vaccine has been proposed by the Infectious Diseases Society of America and the Food and Drug Administration, which underscores the potential persistence of the virus for years to come (Pl. see Ref.). It is crucial to prioritize the needs of at-risk patients, such as those with cancer, during a pandemic or healthcare crisis to minimize the negative impact on their treatment outcomes and overall well-being.
(Ref: E. Mahase, Covid-19: Annual flu-like booster approach may not be appropriate, says expert on infectious disease, British Medical Journal Publishing Group, 2023).”.
- Please acknowledge any potential limitations or biases in the data or sources used for the review
Thanks for your comment
We have added the following paragraph
There are some limitations to this narrative review, including the on-line search. The search is not comprehensive by any means and it is possible that some literature has been inadvertently omitted from this review such as abstracts presented at conferences outside of Europe and America as well as unpublished studies, due to the nature of the search methodology used. During the COVID-19 pandemic, the quality of data collected is impacted by the infrastructure and economic background of each country's healthcare system.
Overall impressions
This paper does not come together in a cohesive way for me and it is unclear what the main point is. I initially thought that the hesitancy to be discussed was going to be about seeking cancer care out of fear of contracting COVID-19. And thus, poor outcomes are expected due to delayed treatment. But then the paper discusses hesitancy for receiving the vaccine. Perhaps this paper is trying to do too much. Or it needs some work to make it fit together better.
Thanks for your comment
We hope the following satisfies your comment
Healthcare deficits in managing cancer patients during the COVID-19 pandemic seem to be largely due to hesitancy or reluctance. The hesitancy was not only from patients avoiding travel to the cancer center because of travel restrictions but was also from physicians restricting patients seen in the healthcare facility. Diagnosis and treatment delays during the pandemic led to health care deficits. Throughout the pandemic, patients with cancer were sidelined because of the more pressing issues in managing COVID-19 infected patients. Healthcare personnel worked offsite in most healthcare facilities in the country. The fear of contracting the disease made healthcare personnel work remotely. One of the reasons for healthcare deficits during the pandemic was the shortage of doctors, nurses, and other healthcare workers in cancer centers. Even though telehealth helped to a certain extent, lack of clinical examination did not help cancer patients. Finally, when vaccines were freely available, providers restricted COVID-19 vaccination to cancer patients. The cancer care during COVID-19 was evolving and response to COVID-19 vaccination for cancer patients was considered suboptimal. Vaccination against COVID-19 was a continuation of the preventive treatment paradigm during the COVID-19 pandemic. Patients with solid tumors who had normal cellular immunity were withheld from receiving the COVID-19 vaccine - further restricting their oncological treatment plan because of the fear of exposure to COVID-19 in chemotherapy infusion centers.
Comments on the Quality of English Language
Minor grammatical errors and inconsistency of formatting (i.e. "COVID-19"), but overall no issues with English language